# DamoFD: Digging into Backbone Design on Face Detection

**Yang Liu [1], Jiankang Deng [2], Fei Wang [1], Lei Shang [1], Xuansong Xie [1], Baigui Sun [1]** [*]
[1]Alibaba Group   [2]Imperial College London
[*]Corresponding Author
{ly261666, steven.wf, sl172005, xingtong.xxs, baigui.sbg}@alibaba-inc.com
jiankangdeng@gmail.com

## Abstract

Face detection (FD) has achieved remarkable success over the past few years, yet, these leaps often arrive when consuming enormous computation costs. Moreover, when considering a realistic situation, *i.e.*, building a lightweight face detector under a computation-scarce scenario, such heavy computation cost limits the application of the face detector. To remedy this, several pioneering works design tiny face detectors through off-the-shelf neural architecture search (NAS) technologies, which are usually applied to the classification task. Thus, the searched architectures are sub-optimal for the face detection task since some design criteria between detection and classification task are different. As a representative, the face detection backbone design needs to guarantee the stage-level detection ability while it is not required for the classification backbone. Furthermore, the detection backbone consumes a vast body of inference budgets in the whole detection framework. Considering the intrinsic design requirement and the virtual importance role of the face detection backbone, we thus ask a critical question: How to employ NAS to search FD-friendly backbone architecture? To cope with this question, we propose a distribution-dependent stage-aware ranking score (DDSAR-Score) to explicitly characterize the stage-level expressivity and identify the individual importance of each stage, thus satisfying the aforementioned design criterion of the FD backbone. Based on our proposed DDSAR-Score, we conduct comprehensive experiments on the challenging Wider Face benchmark dataset and achieve dominant performance across a wide range of compute regimes. In particular, compared to the tiniest face detector SCRFD-0.5GF, our method is +2.5 % better in Average Precision (AP) score when using the same amount of FLOPs. The code is avaliable at `https://github.com/ly19965/EasyFace/tree/master/face_project/face_detection/DamoFD`.

## 1 Introduction

Face detection is a fundamental task in computer vision and plays an important role on various face-related down-streaming applications, *e.g.*, facial expression recognition Zhao et al. (2021), face recognition Deng et al. (1801) and face alignment Ren et al. (2014). In the last decade, we have witnessed tremendous progress on the realm of face detection. However, these leaps arrive only when consuming huge computation cost, such as heavy detection framework in Hambox Liu et al. (2019), TinaFace Zhu et al. (2020), and DSFD Li et al. (2019). Moreover, when building a tiny face detector under a computation-scarce scenario, such heavy computation cost limits the application of face detectors.

It is thus of attracting major research interest on constructing tiny face detectors manually Zhang et al. (2017a); Bazarevsky et al. (2019), which employ SSD Liu et al. (2016) as a basic detection framework and further construct a lightweight network via substituting a manual-designed backbone for SSD feature extractor. However, these methods can only cover a minor range of compute regimes, hindering the application on multiple computation-scarce scenarios. Therefore, follow-up efforts start to pay attention to neural architecture search (NAS) solution, which is a promising direction for developing lightweight face detectors across a wide range of compute regimes. At present,

existing NAS-based FD methods prefer to follow the standard NAS approach to search overall face detection framework, *e.g.*, SPNAS Guo et al. (2019) in BfBox Liu & Tang (2020), RegNet Radosavovic et al. (2020) in SCRFD Guo et al. (2021) and DARTS Liu et al. (2018) in ASFD Zhang et al. (2020a).

However, these methods heavily rely on the power of off-the-shelf NAS approaches while lacking the FD-friendly design, making the searched architectures sub-optimal on the realm of face detection. In this work, we make 3 efforts to develop a novel NAS method for searching lightweight face detectors under various inference budgets automatically, such as inference latency, FLOPs (Floating Point Operations), and model size.

**Search backbone architecture via NAS.** Instead of searching the overall detection framework (unify searching strategy), in this paper, we only use NAS to search the backbone architecture. The rationale is that the detection backbone consumes huge inference costs in most popular detection frameworks Li et al. (2019); Liu et al. (2019); Tang et al. (2018); Deng et al. (2019), indicating the superiority of the detection framework is heavily determined by the backbone architecture. Moreover, the overall detection framework consists of backbone, feature pyramid layer (FPN), and detection head module. As a metric to rank architectures, the accuracy predictor fails to reflect the quality of sampled backbone architecture confidently when adopting unify searching strategy since the backbone is a preceding component of a detection framework.

**Identify the major challenge in the Nas-based backbone design of face detector.** Over the past few years, manual-designed backbones He et al. (2016); Sandler et al. (2018) have achieved significant progress on the task of image classification. Recently, Neural Architecture Search, first introduced by Baker et al. (2016), has progressively emerged as an alternative backbone design method. However, applying NAS technology to the detection backbone is much harder than the classification backbone as the former needs to guarantee the stage-level detection ability while it is not required for the latter. Such task-level discrepancy inevitably results in that existing NAS-based FD methods Liu et al. (2018); Guo et al. (2019); Lin et al. (2021); Radosavovic et al. (2020) are not satisfactory for the detection task. Thus, we can conclude that the major challenge in NAS-based detection backbone design is how to design an accuracy predictor to measure the effectiveness of stage-level detection ability.

**Distribution-dependent stage-aware ranking score.** To solve the aforementioned challenge, we propose a Distribution-dependant Stage-aware Ranking score, termed DDSAR-Score, which acts as an accuracy predictor to measure the quality of sampled backbone architectures from a stage-wise detection ability perspective. Some motivations behind the proposed DDSAR-Score are explained as follows.

(i) In the deep learning theory Pascanu et al. (2013); Hahnloser et al. (2000); Hahnloser & Seung (2001); Pascanu et al. (2013); Bianchini & Scarselli (2014); Telgarsky (2015), the superiority of a ReLU Neural Network (NN) is highly related with its expressivity, *i.e.*, the number of linear regions it can separate its input space into. Based on this property of characterizing the NN representation, we propose a stage-aware ranking score (SAR-Score), which measures the stage-level expressivity in a fair way. Concretely, our stage-aware ranking score has 2 advantages. On the one hand, the computational complexity of the number of the linear region grows exponentially with the increase of ReLU NN's depth, making the expressivity of some backbone architectures hard to obtain. This inevitably hinders the application of using linear regions to characterize ReLU NN's expressivity directly. To deal with this issue, rather than computing the exact linear regions, we adopt the lower bound of the maximal number of linear regions to approximate network expressivity, which can compute the expressivity of any network architecture with extremely low computation cost. Benefiting from this advantage, our method achieves a dominant performance and the searching cost is far less than other sota methods, such as BFbox Liu & Tang (2020), SCRFD Guo et al. (2021), and ASFD Zhang et al. (2020a). On the other hand, due to the chain computation procedure of linear regions, the deeper layer expressivity is more powerful than the shallow one. This indicates that it is no longer confident to reflect the expressivity of shallow stages compared to the deeper ones. To cope with this problem, our stage-aware ranking score makes 3 novel modifications. A detailed explanation of our stage-aware ranking score can be seen in Section 3 and 4.

(ii) After obtaining the stage-level expressivity (SAR-Score), we further propose a Distribution-Dependant Stage-aware Ranking score to linearly combine each stage expressivity according to

individual importance. Inspired by the fact that each stage detection ability should be highly correlated with the ratio of ground-truths matched in the corresponding stage, which is illustrated in Guo et al. (2021), we determine the importance of each stage expressivity (*i.e.*, stage-aware ranking score) according to the prior ground-truths distribution. To this end, the proposed DDSAR-Score is formed.

The rest of the paper is organized as follows. In Section 3, we review how to use linear regions to characterize the ReLU NN's expressivity. In Section 4, we introduce our method and corresponding advantages. Experimental results under different Flops are provided in Section 5. Finally, we conclude the paper in Section 6.

## 2 RELATED WORK

**Face Detection.** Existing Face detection methods focus on solving extreme scale variance through label assignment, scale-level data augmentation strategy, and feature enhancement module. (i) $S^3FD$ Zhang et al. (2017b) introduces a scale compensation anchor matching strategy to help outer faces match enough anchors. Based on a novel observation that some negative anchors equip with significant regression ability, Hambox Liu et al. (2019) introduces an online high-quality anchor mining strategy to mine anchors with extreme regression ability via online information. (ii) Tang et al. (2018) proposes a data-anchor-sampling strategy to increase the ratio of small faces in the training data, which is not a robust solution since it is only excellent at detecting small faces. To solve this, MogFace Liu et al. (2022) analyzes the relationship between the performance of each pyramid layer and the number of ground truths it matches, and further introduces a selective-scale enhancement strategy to maximize the certain pyramid layer detection ability. (iii) SSH Najibi et al. (2017) builds a detection module to increase the receptive field on multiple feature maps. PyramidBox Tang et al. (2018) designs a context-sensitive predict module to enjoy the gain of a wider and deeper network Retinaface Deng et al. (2019) adopts a deformable convolution layer to dynamically increase the context information, improving the face detection performance significantly. Even though tremendous strides have been made by the aforementioned methods efforts, they typically adopt the off-the-shelf backbone architecture (*e.g.*, ResNet-50 He et al. (2016)), causing only a little and plain investigation on the FD-friendly backbone design. In this work, we propose a DDSAR-Score to measure the effectiveness of a backbone architecture in a stage-level perspective, which satisfies the detection backbone design requirement. Then based on the powerful NAS technology, a series of FD-friendly backbones under different inference budgets can be searched efficiently.

**Neural Architecture Search.** Early NAS methods Baker et al. (2016); Liu et al. (2018); Guo et al. (2019) focus on searching backbone architecture on the image recognition task. Typically, they first determine the search space, then search the optimal architecture according to the accuracy predictor. Motivated by the success of NAS on the image recognition task, some researchers begin to put attention to face detection tasks. BFbox Liu & Tang (2020) devises a joint searching strategy to search the backbone and the connection mode of the feature pyramid network simultaneously. ASFD Zhang et al. (2020a) discover the importance of feature enhancement module and search head and feature fusion mode. SCRFD Guo et al. (2021) directly searches the overall detection framework, including backbone, head, and feature fusion mode. All these methods emphasize searching the detection framework in a unified perspective, which damages the quality of searched backbone architecture. In contrast, we propose an effective DDSAR-Score to measure the quality of sampled backbone network from a stage-level perspective, achieving satisfactory computation allocation across different stages.

## 3 PRELIMINARIES

In this paper, we will study how to use the linear region to represent CNN stage-level expressivity impartially, based on which, a robust accuracy predictor can be designed for the subsequent NAS procedure.

First, we formally introduce the conception of the linear region and corresponding background information. Here, the ReLU CNN $\mathcal{N}$ we considered has $L$ hidden convolutional layers and $H$ neurons. Each hidden layer contains one convolutional operator followed by ReLU activation Glorot et al. (2011). For the detailed clarify in $\mathcal{N}$, we denote by $h_0 \times w_0 \times d_0$ the dimension of input neurons $\mathbf{x}^0$.

Analogously, let $h_l \times w_l \times d_l$ be the dimension of output $(\mathbf{x}^l)$ of $l$-th hidden layer , for $1 \leq l \leq L$. Thus, we can view ReLU CNN $\mathcal{N}$ as a piece-wise linear function $\mathcal{F}_{\mathcal{N}}$: $\mathbb{R}^{h_0 \times w_0 \times d_0} \to \mathbb{R}^{h_L \times w_L \times d_L}$;

$$\mathcal{F}_{\mathcal{N}}(\mathbf{x}^0; \theta) = g_L \circ f_L \circ \cdots \circ g_1 \circ f_1(\mathbf{x}^0).$$

where $f_l$ is an affine pre-activation function and $g_l$ is a ReLU activation function $(1 \leq l \leq L)$. The parameter $\theta$ is composed of weight matrices and bias vectors in the $\mathcal{N}$. Following the definition in Pascanu et al. (2013); Montufar et al. (2014); Serra et al. (2018); Brandfonbrener (2018); Hanin & Rolnick (2019a;b), the linear region of a ReLU CNN $\mathcal{N}$ corresponding to $\theta$ can be given by

$$\mathcal{R}_{\theta}^{\mathcal{N}} := \{\mathbf{x}^0 \in \mathbb{R}^{h_0 \times w_0 \times d_0} : g(h(\mathbf{x}^0; \theta)) > 0, \quad \forall h \text{ a neuron in } \mathcal{N}\}$$

where $g(x) = \max\{0, x\}$ is a ReLU function and $h(\mathbf{x}^0; \theta)$ is the pre-activation function of a neuron $h$. Then, we denote by $N_{\mathcal{R}_{\theta}^{\mathcal{N}}}$ the number of the linear regions in $\mathcal{N}$ at $\theta$, i.e., $N_{\mathcal{R}_{\theta}^{\mathcal{N}}} = \#\{\mathcal{R}_{\theta}^{\mathcal{N}} : \mathcal{R}_{\theta}^{\mathcal{N}} \neq \emptyset\}$ [1]. Note that $N_{\mathcal{R}_{\theta}^{\mathcal{N}}}$ has been widely recognized to serve as the expressivity proxy Montufar et al. (2014); Hanin & Rolnick (2019a), when given a neural network $\mathcal{N}$ and corresponding weights $\theta$. Furthermore, let $N_{\mathcal{R}_{max}^{\mathcal{N}}} := \max_{\theta} N_{\mathcal{R}_{\theta}^{\mathcal{N}}}$ be the maximal number of linear regions of $\mathcal{N}$ when $\theta$ ranges over $\mathbb{R}^{\#weights + \#bias}$.

In the remainder of this section, we recall 2 theorems in Xiong et al. (2020), which state the number of linear regions for one-layer ReLU CNN and the lower bound of maximal linear regions for multi-layer ReLU CNNs, respectively.

**Theorem 1 (Theorem 2 from Xiong et al. (2020))** *Assume that $\mathcal{N}$ is a one-layer ReLU CNN with input dimension $h_0 \times w_0 \times d_0$ and hidden layer dimension $h_1 \times w_1 \times d_1$. The $d_1$ filters have the dimension $f_1^{(1)} \times f_1^{(2)} \times d_0$ and the stride $s_1$. Suppose that the parameters $\theta = \{W, B\}$ are drawn from a fixed distribution $\mu$ which has densities with respect to Lebesgue measure in $\mathbb{R}^{\#weights + \#bias}$. Define $I_{\mathcal{N}} = \{(i, j) : 1 \leq i \leq h_1, 1 \leq j \leq w_1\}$ and $S_{\mathcal{N}} = (S_{i,j})_{h_1 \times w_1}$ where*

$$S_{i,j} = \{(a + (i-1)s_1, b + (j-1)s_1, c) : 1 \leq a \leq f_1^{(1)}, 1 \leq b \leq f_1^{(2)}, 1 \leq c \leq d_0\}$$

*for each $(i, j) \in I_{\mathcal{N}}$. Let*

$$K_{\mathcal{N}} := \{(t_{i,j})_{(i,j) \in I_{\mathcal{N}}} : t_{i,j} \in \mathbb{N}, \sum_{(i,j) \in J} t_{i,j} \leq \# \cup_{(i,j) \in J} S_{i,j} \ \forall J \subseteq I_{\mathcal{N}}\}.$$

*Then, the expectation of the number $N_{\mathcal{R}_{\theta}^{\mathcal{N}}}$ of linear regions of $\mathcal{N}$ equals $N_{\mathcal{R}_{max}^{\mathcal{N}}}$:*

$$\mathbb{E}_{\theta \sim \mu}[N_{\mathcal{R}_{\theta}^{\mathcal{N}}}] = N_{\mathcal{R}_{max}^{\mathcal{N}}} = \sum_{(t_{i,j})_{(i,j) \in I_{\mathcal{N}}} \in K_{\mathcal{N}}} \prod_{(i,j) \in I_{\mathcal{N}}} \binom{d_1}{t_{i,j}}. \tag{1}$$

Based on the Theorem 1, Xiong et al. (2020) further derive the bound of $N_{\mathcal{R}_{max}^{\mathcal{N}}}$ on multi-layer CNN:

**Theorem 2 (Theorem 5 from Xiong et al. (2020))** *Suppose that $\mathcal{N}$ is a ReLU CNN with L hidden convolutional layers. The input dimension is $h_0 \times w_0 \times d_0$; the l-th hidden layer has dimension $h_l \times w_l \times d_l$ for $1 \leq l \leq L$; and there are $d_l$ filters with dimension $f_l^{(1)} \times f_l^{(2)} \times d_{l-1}$ and stride $s_l$ in the l-th layer. Assume that $d_l \geq d_0$ for each $1 \leq l \leq L$. Then, the maximal number $N_{\mathcal{R}_{max}^{\mathcal{N}}}$ of linear regions of $\mathcal{N}$ is at least (lower bound)*

$$N_{\mathcal{R}_{max}^{\mathcal{N}}} \geq N_{\mathcal{R}_{max}^{\mathcal{N}'}} \prod_{l=1}^{L-1} \left\lfloor \frac{d_l}{d_0} \right\rfloor^{h_l \times w_l \times d_0}, \tag{2}$$

*where $\mathcal{N}'$ is a one-layer ReLU CNN which has input dimension $h_{L-1} \times w_{L-1} \times d_0$, hidden layer dimension $h_L \times w_L \times d_L$, and $d_L$ filters with dimension $f_L^{(1)} \times f_L^{(2)} \times d_0$ and stride $s_L$. Note that the maximal linear regions $N_{\mathcal{R}_{max}^{\mathcal{N}'}}$ of $\mathcal{N}'$ can be calculated by Eq. 1*

---

[1] Given a set $T$, let $\#T$ be the number of elements in $T$.

## 4 METHODOLOGY

In this section, we first propose a novel SAR-Score to measure the expressivity of different stages and make them comparable, simultaneously. Then, we determine the importance of each stage according to the prior ground-truths distribution, based on which, a DDSAR-Score is proposed to measure the backbone in a unified proxy while preserving the ability to characterize stage-wise expressivity. Finally, combining DDSAR-Score with available NAS technology (*i.e.*, search space design and evolutionary architecture search) completes the searching process on FD backbone architecture.

### 4.1 STAGE-AWARE RANKING SCORE

We first revisit a standard neural architecture search framework, which consists of two key components, architecture generator and accuracy predictor. The former is responsible for generating potential high-quality architecture and the latter is in charge of predicting the corresponding accuracy. Due to the lack of tailored considerations on stage-level representation for existing accuracy predictors, adopting the off-the-shelf NAS technology to design a FD-friendly backbone is often not satisfactory. In this part, we look closely into the lower bound of network expressivity and analyze its issues on characterizing stage-level expressivity, namely unbalanced representation between different stages, huge computation cost on obtaining the lower bound, and non-sensitivity to filter's kernel size.

To handle with the two former weaknesses, we propose a stage-aware expressivity score (SAE-Score). We first clarify some notations. Suppose that $\mathcal{N}$ has a stem and 4 stages $\mathcal{N}_{ci}$ ($i = 2, 3, 4, 5$). For convenience, we term the stem as $\mathcal{N}_{c1}$. The input dimension of $\mathcal{N}$ is $h_0 \times w_0 \times d_0$. For $1 \leq i \leq 5$, $\mathcal{N}_{ci}$ has $L^{ci}$ hidden layers and the input dimension of $\mathcal{N}_{ci}$ is $h_0^{ci} \times w_0^{ci} \times d_0^{ci}$. For $1 \leq l \leq L^{ci}$, the l-th hidden layer ($\mathcal{N}_{ci}^l$) in $ci$ has dimension $h_l^{ci} \times w_l^{ci} \times d_l^{ci}$ and corresponding $d_l^{ci}$ filters have dimension $f_l^{ci} \times f_l^{ci} \times d_{l-1}^{ci}$ with stride $s_l^{ci}$.

By Theorem 2, the maximal number of linear regions of $\mathcal{N}_{ci}$ is at least:

$$N_{\mathcal{R}_{max}^{\mathcal{N}_{ci}}} \geq R_{\mathcal{N}_{ci}} = N_{\mathcal{R}_{max}^{\mathcal{N}_{ci}'}} \prod_{j=1}^{i-1} \prod_{l=1}^{L^{cj}} \left\lfloor \frac{d_l}{d_0} \right\rfloor^{h_l^{cj} \times w_l^{cj} \times d_0} \times \prod_{n=1}^{L^{ci}-1} \left\lfloor \frac{d_n}{d_0} \right\rfloor^{h_n^{ci} \times w_n^{ci} \times d_0} \tag{3}$$

As discussed in Xiong et al. (2020), the lower bound ($R_{\mathcal{N}_{ci}}$) corresponding to the maximal linear region of $\mathcal{N}_{ci}$ can represent the expressivity of $\mathcal{N}_{ci}$. However, two distinct issues occur when using $R_{\mathcal{N}_{ci}}$ to measure the stage-level expressivity directly. (i) As shown in Table 1, we give an illustrator case 1 to unveil that directly using $R_{\mathcal{N}_{ci}}$ (i= $1, 2, 3, 4, 5$) to represent different stage expressivity incurs extremely unbalanced representation (*i.e.*, the lower bound of the maximal linear regions on different stage presents a huge gap). The rationale is that the proof of Theorem 2 is based on the chain calculation, leading the lower bound of the deeper stage increases exponentially relative to the that of the shallow one. This means that it is no longer powerful to reflect the expressivity of shallow stages compared to the deeper ones, when we regard the linear combination of $R_{\mathcal{N}_{ci}}$ ($i = 1, 2, 3, 4, 5$) as the proxy to rank a set of candidate backbone architectures. (ii) The computation cost of $N_{\mathcal{R}_{max}^{\mathcal{N}_{ci}'}}$ is huge, where obtaining the elements in $K_{\mathcal{N}_{ci}'}$ requires crude brute force trial.

**Example 1** *Let $\mathcal{N}$ be a five-layer Relu CNN and the input dimension is $1 \times 1 \times 1$. For $1 \leq l \leq 5$, in the l-th hidden layer, there are $2^l$ filters with dimension $1 \times 1 \times 2^{(l-1)}$ and stride 2. That is, these 5 layers are the corresponding $\mathcal{N}_{c1}, \mathcal{N}_{c2}, \mathcal{N}_{c3}, \mathcal{N}_{c4}, \mathcal{N}_{c5}$. According to Eq. 3, for $1 \leq i \leq 5$, the number of $\mathcal{R}_{max}^{\mathcal{N}_{ci}'}$ (namely $N_{\mathcal{R}_{max}^{\mathcal{N}_{ci}'}}$ ) and $R_{\mathcal{N}_{ci}}$ are reported in Table 1.*

To alleviate this unbalanced representation across different stages and crude computation process of $N_{\mathcal{R}_{max}^{\mathcal{N}_{ci}'}}$, we propose a novel stage-aware expressivity score to measure the stage-wise expressivity

Table 1: the number of $R_{\mathcal{N}_{ci}}$ and $\mathcal{R}_{max}^{\mathcal{N}'_{ci}}$ in Example 1

| | $i=1$ | $i=2$ | $i=3$ | $i=4$ | $i=5$ |
|---|---|---|---|---|---|
| Number of the $\mathcal{R}_{max}^{\mathcal{N}'_{ci}}$ | 3 | 11 | 163 | $3.92 \times 10^4$ | $2.45 \times 10^9$ |
| Number of the $R_{\mathcal{N}_{ci}}$ | 3 | 22 | 652 | $3.13 \times 10^5$ | $3.92 \times 10^{10}$ |

from $\mathcal{N}_{c1}$ to $\mathcal{N}_{c5}$:

$$S_{\mathcal{N}_{ci}}^{sae} = \log(h_{L^{ci}}^{ci} \times w_{L^{ci}}^{ci} \times d_0 \prod_{n=1}^{L^{ci}} \left\lfloor \frac{d_n^{ci}}{d_0} \right\rfloor^{h_n^{ci} \times w_n^{ci} \times d_0}) \tag{4}$$

$$= \log(h_{L^{ci}}^{ci} \times w_{L^{ci}}^{ci} \times d_0) + \sum_{n=1}^{L^{ci}} (h_n^{ci} \times w_n^{ci} \times d_0) \log(\left\lfloor \frac{d_n^{ci}}{d_0} \right\rfloor) \tag{5}$$

The design of our SAE-Score is based on the $R_{\mathcal{N}_{ci}}$ with 2 following novel modifications:

(i) Remove stage-irrelevant item from $R_{\mathcal{N}_{ci}}$. We first review the process of derivation on $\prod_{j=1}^{i-1} \prod_{l=1}^{L^{cj}} \left\lfloor \frac{d_l}{d_0} \right\rfloor$, which represents that the layer from $c1$ to $ck$ ($k = i - 1$) map $\prod_{j=1}^{i-1} \prod_{l=1}^{L^{cj}} \left\lfloor \frac{d_l}{d_0} \right\rfloor$ distinct unit hypercubes in $[0,1]^{h_0 \times w_0 \times d_0}$ into the same hypercube $[0,1]^{h_{L^{ck}}^{ck} \times w_{L^{ck}}^{ck} \times d_0}$. Informally speaking, the value of $\prod_{j=1}^{i-1} \prod_{l=1}^{L^{cj}} \left\lfloor \frac{d_l}{d_0} \right\rfloor$ is only based on the topology structure of the former stage layer while the role of $\mathcal{N}_{ci}$ can be reflected by the remaining term in $R_{\mathcal{N}_{ci}}$. Thus, we can regard $\prod_{j=1}^{i-1} \prod_{l=1}^{L^{cj}} \left\lfloor \frac{d_l}{d_0} \right\rfloor$ as the stage-irrelevant item.

(ii) Re-combination trick: Adding a fully-connected layer with only 1 hidden neuron at the end of $\mathcal{N}_{ci}$.[2] Based on this, we can derive the exact formula of SAE-Score in Eq. 4. The $\prod_{n=1}^{L^{ci}} \left\lfloor \frac{d_n}{d_0} \right\rfloor^{h_n^{ci} \times w_n^{ci} \times d_0}$ in Eq. 4 represents the stage $ci$ map $\prod_{n=1}^{L^{ci}} \left\lfloor \frac{d_n^{ci}}{d_0} \right\rfloor^{h_n^{ci} \times w_n^{ci} \times d_0}$ distinct unit hypercubes in $[0,1]^{h_0 \times w_0 \times d_0}$ into the same hypercube $[0,1]^{h_{L^{ci}}^{ci} \times w_{L^{ci}}^{ci} \times d_0}$. Then by Theorem 3, the new addition layer can divide single hypercube $[0,1]^{h_{L^{ci}}^{ci} \times w_{L^{ci}}^{ci} \times d_0}$ into $h_{L^{ci}}^{ci} \times w_{L^{ci}}^{ci} \times d_0$ linear regions. Finally, the SAE-Score is formed via multiplicating $h_{L^{ci}}^{ci} \times w_{L^{ci}}^{ci} \times d_0$ and $\prod_{n=1}^{L^{ci}} \left\lfloor \frac{d_n^{ci}}{d_0} \right\rfloor^{h_n^{ci} \times w_n^{ci} \times d_0}$.

Thus, compared to the $R_{\mathcal{N}_{ci}}$, our SAR-Score has two advantages. i) The representation across different stages is more balanced because we remove the stage-irrelevant item, whose value increases exponentially with the increase of the stage number, which is demonstrated in Example 1; ii) Eliminating the crude computation process of $N_{\mathcal{R}_{max}^{\mathcal{N}'_{ci}}}$ via re-combination trick.

**Theorem 3 (Proposition 2 from Pascanu et al. (2013))** *Let $\mathcal{N}$ be a one-layer ReLU NN with $n_0$ input neurons and $n_1$ hidden neurons. Then, the maximal number of linear regions of $\mathcal{N}$ is equal to $\sum_{i=0}^{n_0} \binom{n_1}{i}$.*

By taking these advantages of SAE-Score, we can efficiently measure the detection ability across different stages expressivity. However, the SAE-Score is not sensitive to the structure of the filters (*e.g.*, kernel size and stride), resulting in some trivial architectures are searched during the NAS period, *i.e.*, the searched architecture may only contain $3 \times 3$ convolution layer. To solve this, we further propose a filter sensitivity score to rescue the SAE-Score:

$$S_{\mathcal{N}_{ci}}^{fs} = \sum_{l=1}^{L^{ci}} N_{\mathcal{R}_{max}^{\mathcal{N}'_{ci}}} = \sum_{l=1}^{L^{ci}} \sum_{(t_{i,j})_{(i,j) \in I_{\mathcal{N}_{ci}^{l'}}} \in K_{\mathcal{N}_{ci}^{l'}}} \prod_{(i,j) \in I_{\mathcal{N}}} \binom{d'}{t_{i,j}}. \tag{6}$$

Where $I_{\mathcal{N}_{ci}^{l'}} = \{(i,j) : 1 \le i \le h', 1 \le j \le w'\}$ and

$$S_{i,j} = \{(a + (i-1)s_l^{ci}, b + (j-1)s_l^{ci}, c) : 1 \le a \le f_l^{ci}, 1 \le b \le f_l^{ci}, 1 \le c \le d'\}$$

---

[2]This step only involves the calculation of $S_{\mathcal{N}_{ci}}$

for each $(i,j) \in I_{\mathcal{N}_{ci}^{l'}}$. Let

$$K_{\mathcal{N}_{ci}^{l'}} := \{(t_{i,j})_{(i,j) \in I_{\mathcal{N}_{ci}^{l'}}} : t_{i,j} \in \mathbb{N}, \sum_{(i,j) \in J} t_{i,j} \le \# \cup_{(i,j) \in J} S_{i,j} \ \forall J \subseteq I_{\mathcal{N}_{ci}^{l'}}\}.$$

Where $\mathcal{N}_{ci}^{l'}$ is a one-layer ReLU CNN. $d'$ represents the number of fliters which is set to 7. The height, width, and stride of $d'$ filters in $\mathcal{N}_{ci}^{l'}$ are the same as those in $\mathcal{N}_{ci}^{l'}$. For $1 \le l \le L^{ci}$, the input dimension of $\mathcal{N}_{ci}^{l'}$ is $h' \times w' \times d_{l-1}^{ci}$. $h'$ and $w'$ are both set to 1 in this paper. By looking at Theorem 1, the $N_{\mathcal{R}_{max}^{\mathcal{N}_{ci}^{l'}}}$ represents the expectation of the number of $N_{\mathcal{R}_{\theta}^{\mathcal{N}_{ci}^{l'}}}$ of linear regions of $\mathcal{N}_{ci}^{l'}$, when $\theta$ ranges over $\mathbb{R}^{\#weights+\#bias}$. Considering the differences between $\mathcal{N}_{ci}^{l'}$ and $\mathcal{N}_{ci}^{l}$ are the dimension of input neurons, the depth of filters and the number of filters, we can confidently put our trust in that $N_{\mathcal{R}_{max}^{\mathcal{N}_{ci}^{l'}}}$ and $N_{\mathcal{R}_{max}^{\mathcal{N}_{ci}^{l}}}$ have the same role on measuring the expressivity sensitivity to the height, width, and stride of filter. Note that why we use $N_{\mathcal{R}_{max}^{\mathcal{N}_{ci}^{l'}}}$ instead of $N_{\mathcal{R}_{max}^{\mathcal{N}_{ci}^{l}}}$ is that calculating the value of $N_{\mathcal{R}_{max}^{\mathcal{N}_{ci}^{l'}}}$ is extreme quickly and easily, making the Theorem 1 is applicable during the training phase. Finally, by integrating expressivity sensitivity of each layer, the $S_{\mathcal{N}_{ci}}^{fs}$ can be used to measure the expressivity sensitivity of $\mathcal{N}_{ci}$ to the filter.

Finally, we propose a stage-aware ranking score to employ the advantages of stage-aware expressivity score and stage-aware filter sensitivity score, simultaneously:

$$S_{\mathcal{N}_{ci}}^{sar} = S_{\mathcal{N}_{ci}}^{sae} + \alpha \, S_{\mathcal{N}_{ci}}^{fs} \tag{7}$$

Where $\alpha$ represents the importance of $S_{\mathcal{N}_{ci}}^{fs}$. Based on the experimental results, we set $\alpha$ to 0.25 when searching DDSAR-500M models. While for DDSAR-2.5G, DDSAR-10G and DDSAR-34G models, we need to find a suitable value of $\alpha$ and add some constraints into search space. Otherwise, some trivial architectures may be searched.

## 4.2 Distribution-dependent Stage-aware Ranking Score

We have presented how to measure stage-level expressivity by introducing a stage-aware ranking score. Moreover, for the sake of the following NAS procedure, we require a unified proxy to measure the effectiveness of the overall detector architecture. To achieve this, we propose a distribution-dependent stage-aware ranking score, termed DDSAR-Score

$$F_{\mathcal{N}} := \lambda_1 S_{\mathcal{N}_{c1}}^{sar} + \lambda_2 S_{\mathcal{N}_{c2}}^{sar} + ... + \lambda_5 S_{\mathcal{N}_{c5}}^{sar} \tag{8}$$

The weights $\lambda = (\lambda_2, \lambda_3, \lambda_4, \lambda_5)$ are calculated by the following 2 steps: 1) Given a dataset with ground-truths annotation and a Resnet50 backbone; 2) Compute the ratio $(\lambda_2, \lambda_3, \lambda_4, \lambda_5)$ of the all ground-truths matched in the $\mathcal{N}_{c2}$, $\mathcal{N}_{c3}$, $\mathcal{N}_{c4}$, and $\mathcal{N}_{c5}$, respectively. Since anchors are not tiled on the $\mathcal{N}_{c1}$, the weight $\lambda_1$ is set to 0.2 according to the uniform allocation opinion. The motivation behind our DDSAR-Score is that the positive anchor distribution can guide the computation allocation from $\mathcal{N}_{c2}$ to $\mathcal{N}_{c5}$ Guo et al. (2021); Liu et al. (2022), indicating there exists a positive correlation between the positive anchor distribution and stage-level architecture. Thus we adopt the aforementioned steps to determine the value of the weights $\lambda$ under the guidance of ground-truths distribution.

## 4.3 Network Search Space

Following previous worksLin et al. (2021); He et al. (2016); Radosavovic et al. (2020); Sandler et al. (2018), the search space of backbone architecture contains 3 different types of blocks, including residual blocks, bottleneck blocks and moblilenet blocks Sandler et al. (2018). The depth-wise expansion ratio is searched in set $\{1, 2, 4, 6\}$. As mentioned above, the effectiveness of DDSAR-Score is controlled by the Relu CNN network. Thereby, we remove some redundant layers (*i.e.*, Batch Normalization, residual link) when computing DDSAR-Score.

---

**Algorithm 1** Evolutionary Architecture Search

---

**Require:** Search space $\mathcal{A}$, inference budget $B$, max iterations $T$, population size $P$.
**Ensure:** The architecture with the highest DDSAR-Score.
1: $\mathcal{P} :=$ Initialize_population(P, B);
2: **for** $t = 1, 2, \cdots, T$ **do**
3:     Randomly select a network architecture $\mathcal{N}$ from $\mathcal{P}$;
4:     $\mathcal{N}_m = \text{Mutation}(\mathcal{N}, \mathcal{A})$;
5:     **if** $\mathcal{N}_m$ not exceeds inference budget **then**
6:         Calculate DDSAR-Score $F_{\mathcal{N}_m}$ by Eq. 4;
7:         $\mathcal{P} = \mathcal{P} \cup \mathcal{N}_m$;
8:     **end if**
9:     Remove network of the smallest DDSAR-Score if the size of $\mathcal{P}$ exceeds the population size $P$.
10: **end for**
11: Return the architecture with the highest DDSAR-Score in $\mathcal{P}$

---

## 4.4 Evolutionary Architecture Search

In the previous subsection, we presented a novel proxy (DDSAR-Score) to measure the expressivity of the backbone. Then, the subsequent NAS process can be formulated as:

$$a^* = \arg\max_{a \in \mathcal{A}} F_a \tag{9}$$

Where $\mathcal{A}$ is the pre-defined search space. During the architecture search in Eq. 9, we directly adopt Evolutionary Architecture Search Guo et al. (2019). Algorithm 1 describes the detailed searching process. We first construct a search space as illustrated in the above subsection. Then, as described in line 1, we initialize the population $P_0$ according to the inference budget $B$ and population size. After that, at each iteration step t, we randomly select a network architecture $\mathcal{N}$ from $\mathcal{P}$ and mutate it to obtain a child architecture $\mathcal{N}_m$. For the mutation process, we mutate a randomly sampled block in $\mathcal{N}$ to produce a new candidate architecture $\mathcal{N}_m$. If the inference cost of $\mathcal{N}_m$ is less than the inference budget, we add it into the population $\mathcal{P}$. After T iterations, we return the highest DDSAR-Score in the Population $\mathcal{P}$.

## 5 Experiments

### 5.1 Dataset and Implementation Details.

**Training details of NAS phase.** To make a fair comparison with previous works, the training details corresponding to the NAS phase are consistent with Lin et al. (2021). To be concrete, The population size and iteration in Algorithm 1 are set 256 and 96000, respectively. The convolution kernel size is searched from the set $\{3, 5, 7\}$. The searched architecture contains 5 stages, ranging from $\mathcal{N}_{c1}$ to $\mathcal{N}_{c5}$. The inference budgets contain Flops under VAG resolution ($640 \times 480$), inference time, and model parameters. In this paper, we only conduct experiments under Flops constraint. More optimization details, evaluation protocols and Dataset introduction can be seen in Section A.

### 5.2 Ablation Study

Based on the SCRFD-0.5GF detection framework, we conduct ablative experiments to evaluate the effectiveness of the searched backbone architecture through our proposed DDSAR-Score. To conduct a fair comparison with SCRFD-0.5GF, we first compute the Flops (403 Mflops) of the SCRFD-0.5GF backbone under VGA resolution. Thus, the inference budget $B$ in Algorithm 1 is set to 403 Mflops. To this end, we can obtain the searched backbone architecture, which is denoted as DDSAR-0.5GF. Table 5.2 illustrates the results of SCRFD-0.5GF, SCRFD-MobileNet-0.5GF (MobileNet0.25 + SCRFD-0.5GF detection framework), DDSAR-0.5GF (DDSAR-0.4GF + SCRFD-0.5GF detection framework). By directly employing the searched DDSAR-0.5GF backbone on the SCRFD-0.5F detection framework, our proposed method achieves a great improvement of 2.49% on the challenging Wider Face hard subset. Besides, the searching cost of our method is

Table 2: Results of the state-of-the-art face detection methods on Wider Face validation dataset. * denotes the result is obtained from scrfd open-source code.

| Method | Easy | Medium | Hard | Params(M) | Flops(G) |
|---|---|---|---|---|---|
| DSFD Li et al. (2019) | 94.29 | 91.47 | 71.39 | 120.06 | 259.55 |
| FaceBoxes Zhang et al. (2017a) | 76.17 | 57.17 | 24.18 | 1.01 | 0.275 |
| RetinaFace Deng et al. (2019) | 94.92 | 91.90 | 64.17 | 29.50 | 37.59 |
| HAMBox Liu et al. (2019) | 95.27 | 93.76 | 76.75 | 30.24 | 43.28 |
| TinaFace Zhu et al. (2020) | 95.61 | 94.25 | 81.43 | 37.98 | 172.95 |
| SCRFD-34GF* | 95.81 | 94.92 | 85.39 | 9.80 | 34.13 |
| DDSAR-34GF | 95.63 | 94.80 | 86.08 | 6.24 | 34.03 |
| SCRFD-10GF* | 94.89 | 93.72 | 82.69 | 3.86 | 9.98 |
| DDSAR-10GF | 95.14 | 94.29 | 84.07 | 1.27 | 9.74 |
| SCRFD-2.5GF* | 93.75 | 92.01 | 77.48 | 0.67 | 2.53 |
| DDSAR-2.5GF | 92.82 | 91.48 | 78.70 | 0.44 | 2.46 |
| SCRFD-0.5GF | 90.57 | 88.12 | 68.51 | 0.57 | 0.508 |
| DDSAR-0.5GF | 90.32 | 88.36 | 71.03 | 0.26 | 0.520 |

only 1 GPU hours, which is far less than counterparts (100+ GPU hours). Such rare searching costs and higher detection performance consistently demonstrate the superiority and great potential of our proposed DDSAR-Score. Due to the limitation of page size, the detail structure of searched backbone under different Flops is placed on Section A.

Table 3: Ablation studies for DDSAR-Score on the Wider Face validation dataset.

| Method | Easy | Medium | Hard | Params(M) | Flops(G) | Seaching Cost (GPU Hour) |
|---|---|---|---|---|---|---|
| SCRFD-MobileNet-0.5GF | 90.38 | 87.05 | 66.68 | 0.37 | 0.507 | 0 |
| SCRFD-0.5GF | 90.57 | 88.12 | 68.51 | 0.57 | 0.508 | 100+ |
| DDSAR-0.5GF | 90.32 | 88.36 | 71.03 | 0.26 | 0.496 | 1.0 |

## 5.3 Comparison with State of the Art

Under different inference budgets, we can search a FD backbone family. Then, integrating them into the SCRFD detection framework, the DDSAR family is formed, including DDSAR-0.5GF, DDSAR-2.5GF, DDSAR-10GF, and DDSAR-34GF. As such, our DDSAR family can be compared with existing state-of-the-art methods in a fair way, *e.g.*, SCRFD family, DSFD Li et al. (2019), RetinFace Deng et al. (2019), TinaFace Zhu et al. (2020) and FaceBoxes Zhang et al. (2017a). As shown in Table 5.1, our DDSAR family achieves the best performance under different compute regimes. This demonstrates that whatever manual-designed or existing NAS-based backbones, the corresponding backbone representation is inferior to ours. In our opinion, considering the manual-designed backbones are derived from image recognition tasks, such significant improvements reveal the importance and effectiveness of designing a backbone on the face detection realm. While comparing with SCRFD family models, our DDSAR-Score continuously takes the stage-level expressivity and prior ground-truths distribution into account, thus a FD-friendly backbones can be searched.

## 6 Conclusion

In this paper, we aim to employ NAS to search FD-friendly backbone. First, we discover that the off-the-shelf NAS technology fails to consider stage-wise detection ability, making the searched backbone sub-optimal on the face detection realm. Secondly, we propose a stage-aware expressivity score to characterize stage-level detection ability explicitly. Thirdly, we further propose a DDSAR-Score to linearly combine each stage expressivity (SAR-score) according to the prior ground-truths distribution. Extensive experiments on the authoritative and challenging Wider Face dataset demonstrate the superiority of our approach.

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

# A APPENDIX

## A.1 TRAINING ON THE SEARCHED NETWORK

To align with existing state-of-the-art SCRFD Guo et al. (2021), we apply our searched backbone on the SCRFD-family models by substituting the original backbone, including SCRFD-0.5GF, SCRFD-2.5GF, SCRFD-10GF, SCRFD-34GF. For the anchor setting, we tile anchors of $\{16, 32\}$, $\{64, 128\}$, $\{256, 512\}$ on the feature maps with strides 8, 16, and 32, respectively. In the detection head module, we adopt weight sharing Deng et al. (2019) and group normalization Wu & He (2018). During the label assignment phase, we employ Adaptive Training Sample Selection Zhang et al. (2020b) to divide positive and negative anchors. For data augmentation strategy, we adopt sample redistribution strategy Guo et al. (2021), which first randomly crops the square patches from the image with a random size from $[0.35, 0.45, 0.6, 0.8, 1.0, 1.2, 1.4, 1.6, 1.8, 2.0]$, and then the sampled square patches are resized to $640 \times 640$. Color distortion and random horizontal flipping are also applied during the data augmentation phase. The optimization objectives of classification and localization branches are Generalised Focal Loss and DIoU Loss, respectively. For the optimization details, We adopt the SGD optimizer (momentum 0.9, weight decay 5e-4) with a batch size of $8 \times 4$ and train on four Tesla V100s. The initial learning rate is set to 1e-5, linearly warming up to 1e-2 within the first 3 epochs. All these mentioned training details are the same as SCRFD Guo et al. (2021), under this setting, we can fairly verify the effectiveness of our proposed method.

## A.2 EVALUATION PROTOCOLS.

Following the previous works Deng et al. (2019); Guo et al. (2021), we adopt the average precision score to serve as the evaluation metric. We adopt a single-scale testing strategy to evaluate the experiment result. Firstly, we feed the image with VGA resolution ($640 \times 480$) into the detector and then get the top-5000 highest confidence bounding boxes. Then, the Non-maximum Suppression is applied with the IoU threshold 0.45 to get the top 750 confident detection scores and related bounding boxes.

## A.3 DATASET.

In this paper, all experiments are conducted on the authoritative and challenging Wider Face Yang et al. (2016) dataset. This famous dataset contains 32203 images and 393703 faces, which are divided into 61 event classes with a high degree of variability in scale, pose, and occlusion. In each event, images are randomly separated into training (50%), validation (10%), and test (40%) sets. According to the detection result on the Wider Face validation and test sets, all 393703 faces are classified into Easy, Medium, and Hard subsets.

Table 4: Searched backbone architecture of DDSAR-0.5GF

| block | kernel | in | out | stride | bottleneck | # layers | level |
|-------|--------|-----|-----|--------|------------|----------|-------|
| Conv | 3 | 3 | 32 | 2 | - | 1 | C1 |
| MB | 3 | 32 | 32 | 2 | 8 | 1 | C2 |
| MB | 7 | 32 | 64 | 2 | 40 | 1 | C3 |
| MB | 7 | 64 | 120 | 2 | 40 | 2 | C4 |
| MB | 5 | 120 | 160 | 2 | 120 | 1 | C5 |

Table 5: Searched backbone architecture of DDSAR-2.5G

| block | kernel | in | out | stride | bottleneck | # layers | level |
|-------|--------|-----|-----|--------|------------|----------|-------|
| Conv | 3 | 3 | 24 | 2 | - | 2 | C1 |
| Res | 3 | 24 | 32 | 2 | 24 | 4 | C2 |
| Res | 3 | 32 | 56 | 2 | 16 | 4 | C3 |
| Res | 5 | 56 | 88 | 2 | 8 | 5 | C4 |
| Btn | 5 | 88 | 128 | 2 | 8 | 4 | C5 |

Table 6: Searched backbone architecture of DDSAR-10G

| block | kernel | in | out | stride | bottleneck | # layers | level |
|-------|--------|-----|-----|--------|------------|----------|-------|
| Conv | 3 | 3 | 56 | 2 | - | 2 | C1 |
| Btn | 3 | 56 | 80 | 2 | 40 | 5 | C2 |
| Btn | 5 | 80 | 112 | 2 | 32 | 5 | C3 |
| Btn | 3 | 112 | 168 | 2 | 16 | 4 | C4 |
| Res | 7 | 168 | 256 | 2 | 8 | 1 | C5 |

Table 7: Searched backbone architecture of DDSAR-34G

| block | kernel | in | out | stride | bottleneck | # layers | level |
|-------|--------|-----|-----|--------|------------|----------|-------|
| Conv | 3 | 3 | 56 | 2 | - | 2 | C1 |
| Btn | 3 | 56 | 224 | 2 | 80 | 5 | C2 |
| Btn | 3 | 224 | 488 | 2 | 40 | 6 | C3 |
| Btn | 3 | 488 | 624 | 2 | 72 | 4 | C4 |
| Btn | 7 | 624 | 816 | 2 | 24 | 3 | C5 |

## A.4 DETAIL STRUCTURE

Following the design in Lin et al. (2021); Sun et al. (2022), We list detail structure in Tab. 4,5,6,7. The 'block' column is the block type. 'Conv' is the standard convolution layer followed by BN and RELU. 'Res' is the residual block used in ResNet-18. 'Btn' is the residual bottleneck block used in ResNet-50. 'MB' is the MobileBlock used in MobileNet and EfficientNet. To be consistent with 'Btn' block, each 'MB' block is stacked by two MobileBlocks. That is, the kxk full convolutional layer in 'Btn' block is replaced by depth-wise convolution in 'MB' block. 'kernel' is the kernel size of kxk convolution layer in each block. 'in', 'out' and 'bottleneck' are numbers of input channels, output channels and bottleneck channels respectively. 'stride' is the stride of current block. ' layers' is the number of duplication of current block type.

