# OpenReview forum: "DamoFD: Digging into Backbone Design on Face Detection"
_ICLR.cc/2023/Conference — ICLR 2023 poster_

### Official Review · Reviewer_8dSd · 2022-10-20

**Confidence:** 4
**Correctness:** 4
**Technical Novelty And Significance:** 3
**Empirical Novelty And Significance:** 3
**Recommendation:** 8

**Clarity, Quality, Novelty And Reproducibility:**

This manuscript is written and organized well. The proposed method is interesting and novel. However, the experiments should be enriched to convince the readers.

**Strength And Weaknesses:**

Strength:
The proposed method is reasonable and feasible. Also, the corresponding theoretical proofs are sufficient. From the reported experimental results, the usefulness of DDSAR-Score can be confirmed.
Weaknesses:
1.	The literature review related to NAS should be enriched.
2.	The explanation of formulae should be improved. For example, what does $\circ $ mean in the definition of ${{\mathcal{F}}_{\mathcal{N}}}\left( {{x}^{0}};\theta  \right)$?
3.	The strategy for selecting weights in Section 4.2 is described as shallow.
4.	Is the selection of the evolutionary architecture search algorithm a limitation of your work? In other words, can any other search algorithms work?
5.	How do you define the network search space?
6.	Testing data could be enriched.
7.	More methods should be chosen to testify to your method, especially the latest ones.
8.	Reference should be updated. Most of the cited literature in the current version was published before 2020, which would confuse the readers that your research topic is not essential anymore.


**Summary Of The Paper:**

This paper proposes a distribution-dependent stage-aware ranking score (DDSAR-Score) for efficient face detection (FD), which can explicitly characterize the stage-level expressivity and identify the individual importance of each stage, thus satisfying the model structure design criterion of the FD backbone. As an accuracy predictor, DDSAR-Score measures the quality of sampled backbone architectures stage-wise, which is beneficial to apply the neural architecture search (NAS) technique to FD.

**Summary Of The Review:**

Based on the above comments, my decision is Accept.

---

> ### Author Response · Authors · 2022-11-10
> **Author's Response to Reviewer 8dSd**
>
> Thanks for your appreciation of our paper. We have definitely revised our manuscript according to your high-quality advices, e.g., enrich literature review related to NAS [1,2,3,4], improve the explanation of formulae, add more methods to verify the effectiveness of our method (i.e., ASFD, DSFD, pyramidbox and CRFace), update reference. Answers to remaining points are below:
>
> **Q1**: How do you define the network search space?
>
> **A1**: Following previous works [5], the search space of backbone architecture contains 3 different types of blocks, including residual blocks, bottleneck blocks and moblilenet blocks. The depth-wise expansion ratio is searched in set $\{1, 2, 4, 6\}$. As mentioned in the submitted manuscript, the effectiveness of DDSAR-Score is controlled by the Relu CNN network. Thereby, we remove some redundant layers (i.e., Batch Normalization, residual link) when computing DDSAR-Score and then added back in training and testing stages.
> These structures will not significantly affect the representation power of networks. For example, BN layer can be
> merged into convolutional kernel via kernel fusion. Self attention linearly combines existing feature maps and hence spans the same subspace.
>
> **Q2**:  Is the selection of the evolutionary architecture search algorithm a limitation of your work? In other words, can any other search algorithms work?
>
> **A2**: Yes. This work focuses on investigate the design of accuracy predictor while neglecting its compatibility with different types of search algorithms.
>
> [1] Elsken T, Metzen J H, Hutter F. Neural architecture search: A survey[J]. The Journal of Machine Learning Research, 2019, 20(1): 1997-2017.
>
> [2] Liu Y, Sun Y, Xue B, et al. A survey on evolutionary neural architecture search[J]. IEEE transactions on neural networks and learning systems, 2021.
>
> [3] Jaafra Y, Laurent J L, Deruyver A, et al. Reinforcement learning for neural architecture search: A review[J]. Image and Vision Computing, 2019, 89: 57-66.
>
> [4] Chitty-Venkata K T, Emani M, Vishwanath V, et al. Neural Architecture Search for Transformers: A Survey[J]. IEEE Access, 2022.
>
> [5] Lin M, Wang P, Sun Z, et al. Zen-nas: A zero-shot nas for high-performance image recognition[C]//Proceedings of the IEEE/CVF International Conference on Computer Vision. 2021: 347-356.

---

### Official Review · Reviewer_9WwS · 2022-10-20

**Confidence:** 4
**Correctness:** 3
**Technical Novelty And Significance:** 2
**Empirical Novelty And Significance:** 2
**Recommendation:** 6

**Clarity, Quality, Novelty And Reproducibility:**

The major contribution is to derive the ranking score for \emph{multi-stage} networks from the expressivity of ReLU CNNs, which is modest novel. The technical approach is sound. The proposed method seems not that easy to reproduce by other researchers. The paper is clearly written.

**Strength And Weaknesses:**

Face detection has been extensively studied in decades with many commercial face detectors running on embedded systems. Search for a lightweight backbone network for face detection using NAS is an interesting and well-motivated topic. The proposed DDSAR-Score reasonably integrates the expressivity of multi-stage ReLU CNNs, which is somewhat novel and empirically works well in an evolutionary architecture search.

The proposed stage-aware ranking score approximates the expressivity by the lower bound of the maximal number of linear regions, which is fine. The major concern is the gap between the expressivity of ReLU CNN and the face detection performance, e.g. detection rate and false positive rate. Please discuss more about the relation between the network expressivity and detection performance.

Another concern is the generality of the proposed DDSAR-Score, which is derived from the expressivity of ReLU CNNs in previous works. How to extend DDSAR-Score to other network architectures?


**Summary Of The Paper:**

The paper studied how to search for a lightweight backbone network for face detection using NAS. The key is how to develop stage-aware accuracy predictors of face detection performance for different network architectures. Thus, the paper derived a new stage-aware ranking score from the expressivity of ReLU CNNs, i.e., the lower bound of the maximal number of linear regions, from previous works (e.g., Xiong et al. 2020). Specifically, the paper proposed 1) a stage-aware expressivity score (Eq.4) by removing stage-irrelevant items and adding a fully-connected layer to combine multiple layers; and 2) calibrating the weights (Eq.8) to combine the ranking scores across multiple stages by a dataset with groundtruths and a Resnet50 backbone. The proposed ranking scores, i.e., DDSAR-Score, is integrated in an evolutionary architecture search. This method has been evaluated on the Wider Face dataset and compared favorably with recent NAS methods for face detection.

**Summary Of The Review:**

Overall, this is a descent paper addressing a well-motivated problem. The proposed stage-aware expressivity score is new and works well. Well, face detectors have been integrated in many face recognition systems on embedded platforms. Usually, the face detector is not the bottleneck for either model size or computation.

---

> ### Author Response · Authors · 2022-11-10
> **Author's Response to Reviewer 9WwS**
>
> We thank the reviewer for the valuable comments. We are glad that you like our work, feel that our paper is interesting and well-motivated.  Below, we address your comments on weaknesses point-by-point.
>
> **Q1**: Please discuss more about the relation between the network expressivity and detection performance, e.g. detection rate and false positive rate.
>
> **A1**: Thanks for your comment. We add a table to reveal the relation between the network expressivity and detection performance (i.e., AP score on easy, medium, and hard sets). The inference budget B is set to 403 Mflops.
> Apparently, with the increasing of DDSAR-score, the AP performance on hard set also increases synchronously, indicating the positive correlation between the network expressivity and the detection performance.
> Note that the gt distribution of the training set is consistent with that of the hard set rather than the easy and medium sets, thus the positive correlation phenomenon is not obvious in the latter two sets. Additionally, the hard set contains the entire test data, namely including all test data in the easy and medium set.
>
> | DDSAR | Easy | Medium | Hard |
> | :----: | :----: |:----: |:----: |
> |  92.14 | 90.79 | 88.43 | 71.00 |
> | 90.72 | 91.10 | 88.53 | 70.47 |
> | 89.23 | 90.87 | 88.14 |  70.13 |
>
> **Q2**: Another concern is the generality of the proposed DDSAR-Score, which is derived from the expressivity of ReLU CNNs in previous works. How to extend DDSAR-Score to other network architectures?
>
> **A2**: Yes,  DDSAR-Score can only be used in ReLU CNNs architecture. To satisfy this constraint condition, our search space only contains ReLU CNN architecture while the unsatisfactory layers (i.e., Batch Normalization, Self-attention Block) are removed when computing the DDSAR-Score and then added back in the training and testing stages.
> These structures will not significantly affect the representation power of networks. For example, BN layer can be
> merged into a convolutional kernel via kernel fusion. Self-attention linearly combines existing feature maps and  hence spans the same subspace.

---

### Official Review · Reviewer_XsaM · 2022-10-24

**Confidence:** 4
**Correctness:** 3
**Technical Novelty And Significance:** 3
**Empirical Novelty And Significance:** 2
**Recommendation:** 6

**Clarity, Quality, Novelty And Reproducibility:**

In general, the paper is well-motivated. The idea of measuring stage-wise expressivity for NAS is interesting.  Experimental results show improvements on face detection benchmark.


**Strength And Weaknesses:**

STRENGTH
- The paper is well-motivated.
- The idea of measuring stage-wise expressivity for NAS is interesting.
- Experimental results show improvements on face detection benchmark.

WEAKNESS
1. It is unclear how the authors come up with the configuration for $N^{l’}_{ci}$.
An ablation study for selecting these configurations is recommended.

2. The ablation study for the contributions of each component, i.e. $S^{sae}\_{N\_{ci}}$, $S^{fs}\_{N\_{ci}}$ is missing.
The authors should discuss the contributions of these components to show the advantages of each proposed term.

3. In Section 4.2, Resnet50 backbone is adopted as a reference for ratios ($\lambda_2,\lambda_3,\lambda_4, \lambda_5$).
How these numbers changes if different reference backbones are adopted? How do these values affect the obtained FD backbone?

4. The performance improvements of DDSAR are marginal.
Why does DDSAR improve on hard set but not improve on Easy and Medium sets?

5. The obtained architectures should be presented and discussed.


**Summary Of The Paper:**

In this paper, the authors introduce a distribution-dependent stage-aware ranking score (DDSAR-Score) for NAS to search FD-friendly backbone architectures.
The proposed DDSAR-score aims at estimating the stage-level expressivity and identify the importance of each stage which can be used for NAS.
Several modifications are proposed such as: stage-irrelevant term removal, re-combination trick and filter sensitivity score.
DDSAR score shows its potential in improving the Face detection on WiderFace benchmark.


**Summary Of The Review:**

In general, the paper is well-motivated. Experiments also show its improvements on FD benchmark.
However, There are some ablation studies on the contributions of the proposed terms are missing as mentioned in Weakness Section.

---

> ### Author Response · Authors · 2022-11-10
> **Author's Response to Reviewer XsaM (Part-1 out of 2)**
>
> We thank the reviewer for the constructive comments and for taking the time to thoroughly read this paper. We are glad that you think our work is well-motivated and interesting. Below, we address your comments on weaknesses point-by-point.
>
> **Q1**: It is unclear how the authors come up with the configuration for $N_{ci}^{l'}$. An ablation study for selecting these configurations is recommended.
>
> **A1**: This configuration for $N_{ci}^{l'}$ is the most simple configuration that satisfies the definition of $S_{N_{ci}}^{fs}$, which is illustrated on Equ.6 in the submitted manuscript.
> Although enlarging the architecture of $N_{ci}^{l'}$ can also satisfy the definition of $S_{N_{ci}}^{fs}$, the computation cost will increase exponentially according to its computation process (described in Theorem 1). We also provide an illustrator example in Example 1 to display this complicated computation process.
>
> **Q2**: The ablation study for the contributions of each component, i.e.
> , $S_{N_{ci}}^{sae}$, $S_{N_{ci}}^{fs}$ is missing.
>
> **A2**: Thanks for the suggestion. We add an ablation study about SAR-Score / FS-Score and  the results are listed as below. The inference budget B is set to 403 Mflops. Besides, we also list the detail of searched architectures with SAR-Score and SAR-Score + FS-Score proxies, respectively. The 'block' column is the block type. 'Conv' is the standard convolution layer followed by BN and RELU. 'ResBlock' is the residual bottleneck block used in ResNet-50 and is stacked by 3 Blocks in our design. Mobile block is adopted in this experiment setting. 'kernel' is the kernel size of kxk convolution layer in each block. 'in', 'out' and 'bottleneck' are numbers of input channels, output channels and bottleneck channels respectively. 'stride' is the stride of current block. '$\#$ layers' is the number of duplication of current block type.
> As revealed from the below table, the searched architectures with SAR-Score proxy only contain 3x3 'Conv' , falling into the trivial architecture search result, which is also the motivation of why we design FS-Score to avoid trivial solution.
>
>
> - Ablation study of SAR-Score and FS-Score on Wider Face dataset,  $\alpha=0.25$.
> | Proxy | Easy | Medium | Hard |
> | :---- | :----: |:----: |:----: |
> |SAR-Score | 89.87 |  87.24 | 66.14 |
> | SAR-Score + FS-Score | 90.79 | 88.43 | 71.00 |
>
>
> - Detailed architecture searched by "SAR-Score + FS-Score" proxy, $\alpha=0.25$.
>
> | block | kernel | in | out | bottleneck | stride | # layers | level |
> | :----: | :----: |:----: |:----: |:----: |:----: |:----: |:----: |
> | Conv | 3 | 3 | 32 | - | 2 | 1 | C1|
> | MB | 3 | 32 | 32 | 8 | 2 | 1 | C2|
> | MB | 5 | 32 | 64 | 40 | 2 | 1 | C3|
> | MB | 3 | 64 | 120 |  40 | 2 | 2 | C4|
> | MB | 5 | 120 | 160 | 120 | 2 | 1 | C5|
>
> **Q3**: How these numbers change if different reference backbones are adopted? How do these values affect the obtained FD backbone?
>
> **A3**: (i) We adopt Resnet50 as a typical example to illustrate the detailed computation procedure, where the concrete backbone architecture is not involved in this step. Thus, it will not affect the value of $(\lambda_2, \lambda_3, \lambda_4, \lambda_5)$  when varying the architecture of reference backbones.
> (ii) If we remove $(\lambda_2, \lambda_3, \lambda_4, \lambda_5)$, the obtained FD backbone fails to measure the importance of each stage and is not sensitive to the dataset varying, thus a sub-optimal architecture is searched.
> While for our proposed DDSAR-score, it determines the role of each stage through prior gt distribution, which mitigates the aforementioned problems.

---

> > ### Author Response · Authors · 2022-11-10
> > **Author's Response to Reviewer XsaM (Part-2 out of 2)**
> >
> > **Q4**: The performance improvements of DDSAR are marginal. Why does DDSAR improve on hard set but not improve on Easy and Medium sets?
> >
> > **A4**: Thanks for your comment. In our opinion, this descent trending on the easy and hard subset due to the introduction of our proposed DDSAR-Score, which determines the importance of each stage via prior gt distribution.
> > Considering that the scale of almost 90\% faces in the Wider Face dataset is less than 66, the importance of the deep stage (i.e., c4,c5) is relatively less critical. Based on this prior gt distribution, the proposed DDSAR-Score enlarges the importance of shallow stages.
> > Thus, our DDSAR decreases on the easy and hard subset since most faces in them are required to detect on the deep stages.
> >
> > **Q5**: The obtained architectures should be presented and discussed.
> >
> > **A5**: In A2, we have illustrated the detail architecture of our proposed SCRFD-DDSAR-0.5GF. Compared to light-weight backbone MobileNet0.25, the most obvious difference is that our SCRFD-DDSAR-0.5GF assigns more computation cost on shallow stages.
> > Intuitively, this computation assignment is reasonable since the tiny faces are often tiled on the shallow stage and the challenging problem in face detection is how to solve tiny face detection effectively.

---

### Official Review · Reviewer_g2gn · 2022-10-25

**Confidence:** 4
**Correctness:** 3
**Technical Novelty And Significance:** 3
**Empirical Novelty And Significance:** 3
**Recommendation:** 6

**Clarity, Quality, Novelty And Reproducibility:**

The work is novel and well-illustrated. However, more experiments might be included to support their results.

**Strength And Weaknesses:**

Strengths:
1, The main theories are well explained, and the math functions are correctly organized.
2, The proposed DDSAR-score is novel and makes the NAS FD-friendly.

However, I have some concerns:
1, In section 5.3, it is unable to find Table 5.1 in the main manuscript.
2, There is no ablation study about the proposed stage-aware expressivity score and stage-aware filter sensitivity score. It is better to have some experiments to illustrate their contributions.
3, In Table2, the performance of SCRFD-DDSAR-34GF decreases on the easy and hard subset. It might be better to add some comments on this.


**Summary Of The Paper:**

Since the existing neural architecture search (NAS) technologies cannot explore the stage-level detection ability required by face detection tasks, this paper proposed a distribution-dependent stage-aware ranking score to evaluate the expressivity of the backbone. Based on the proxy score, they conduct the NAS on SCRFD family detectors and achieve better performance on both accuracy and inference speed. SOTA results are reported in the paper.

**Summary Of The Review:**

The proposed method makes the NAS work better in face detection, which is relatively novel. However, insufficient experiments affect the clarity of the manuscript.

---

> ### Author Response · Authors · 2022-11-10
> **Author's Response to Reviewer g2gn**
>
> We thank the reviewer for the constructive comments and for taking the time to thoroughly read this paper. We are glad that you think our work is novel and the main theories are well explained . Below, we address your comments on weaknesses point-by-point.
>
> **Q1**: In section 5.3, it is unable to find Table 5.1 in the main manuscript.*
>
> **A1**: We thank the reviewer for this suggestion. We fix this problem in the revised version through adding reference on the correct table, namely Table 2 in the submitted manuscript.
>
> **Q2**: There is no ablation study about the proposed stage-aware expressivity score and stage-aware filter sensitivity score.
>
> **A2**: Thanks for the suggestion. We add an ablation study about SAR-Score / FS-Score and  the results are listed as below. The inference budget B is set to 403 Mflops. Besides, we also list the detail of searched architectures with SAR-Score and SAR-Score + FS-Score proxies, respectively. The 'block' column is the block type. 'Conv' is the standard convolution layer followed by BN and RELU. 'ResBlock' is the residual bottleneck block used in ResNet-50 and is stacked by 3 Blocks in our design. Mobile block is adopted in this experiment setting. 'kernel' is the kernel size of kxk convolution layer in each block. 'in', 'out' and 'bottleneck' are numbers of input channels, output channels and bottleneck channels respectively. 'stride' is the stride of current block. '$\#$ layers' is the number of duplication of current block type.
> As revealed from the below table, the searched architectures with SAR-Score proxy only contain 3x3 'Conv' , falling into the trivial architecture search result, which is also the motivation of why we design FS-Score to avoid trivial solution.
>
>
> - Ablation study of SAR-Score and FS-Score on Wider Face dataset,  $\alpha=0.25$.
> | Proxy | Easy | Medium | Hard |
> | :---- | :----: |:----: |:----: |
> |SAR-Score | 89.87 |  87.24 | 66.14 |
> | SAR-Score + FS-Score | 90.79 | 88.43 | 71.00 |
>
> - Detail architecture searched by "SAR-Score + FS-Score" proxy, $\alpha=0.25$.
>
> | block | kernel | in | out | bottleneck | stride | # layers | level |
> | :----: | :----: |:----: |:----: |:----: |:----: |:----: |:----: |
> | Conv | 3 | 3 | 32 | - | 2 | 1 | C1|
> | MB | 3 | 32 | 32 | 8 | 2 | 1 | C2|
> | MB | 5 | 32 | 64 | 40 | 2 | 1 | C3|
> | MB | 3 | 64 | 120 |  40 | 2 | 2 | C4|
> | MB | 5 | 120 | 160 | 120 | 2 | 1 | C5|
>
>
> **Q3**: In Table2, the performance of SCRFD-DDSAR-34GF decreases on the easy and hard subset.
>
> **A3**: Thanks for your comment. In our opinion, this descent trending on the easy and hard subset due to the introduction of our proposed DDSAR-Score, which determines the importance of each stage via prior gt distribution.
> Considering that the scale of almost 90\% faces in the Wider Face dataset is less than 66, the importance of the deep stage (i.e., c4,c5) is relatively less critical. Based on this prior gt distribution, the proposed DDSAR-Score enlarges the importance of shallow stages.
> Thus, the performance of SCRFD-DDSAR-34GF decreases on the easy and hard subset since most faces in them are required to detect on the deep stages.

---

### Decision · Program_Chairs · 2023-01-20

**Decision:**

Accept: poster

**Justification For Why Not Higher Score:**

This work proposes a new solution to an important problem. However, the reviewers also share some concerns on (1) the performance advantage is not very significant compared with existing works and (2) some details and ablation experiments are lacking, and also in-depth analysis on the method is lacking. These limit its impact in the community.

**Justification For Why Not Lower Score:**

All the reviewers give accept ratings. This work proposes a new solution to an important problem. Though the performance improvement is marginal, its method and resulted models should be valuable for the face detection community.

**Metareview: Summary, Strengths And Weaknesses:**

This paper explores NAS approaches to search for a lightweight face detection backbone. Though there is plenty of NAS methods for classification models, searching for a face detector backbone is still rare due to the challenges of the multi-stage nature. This paper proposes a new distribution-dependent stage-aware ranking score to enable searching for face detector backbones. Experiment results demonstrate the yielded backbone works well.

Strength:

- The studied problem, i.e., how to design light-weight face detectors, is important.
- The proposed method is novel.
- The yielded backbone works empirically well in the experiments.

Weakness:
- Some technical details can be clearer.
- The performance over existing models are not very significant.

**Note From Pc:**

if the above contains the word "oral" or "spotlight" please see: "oral" presentation means -> notable-top-5% and "spotlight" means -> notable-top-25%. As stated in our emails, we are disassociating presentation type from AC recommendations